# Monitoring trends and differences in COVID-19 case-fatality rates using decomposition methods: Contributions of age structure and age-specific fatality

Christian Dudel[1]*, Tim Riffe[1], Enrique Acosta[1], Alyson van Raalte[1], Cosmo Strozza[2,3], Mikko Myrskylä[1,4]

1 Max Planck Institute for Demographic Research, Rostock, Germany, 2 Sapienza University of Rome, Rome, Italy, 3 Interdisciplinary Centre on Population Dynamics, University of Southern Denmark, Odense, Denmark, 4 Population Research Unit, University of Helsinki, Helsinki, Finland

* dudel@demogr.mpg.de

**Data Availability Statement:** All code and data is available from a repository on the Open Science Framework, DOI 10.17605/OSF.IO/VDGWT.

## Abstract

The population-level case-fatality rate (CFR) associated with COVID-19 varies substantially, both across countries at any given time and within countries over time. We analyze the contribution of two key determinants of the variation in the observed CFR: the age-structure of diagnosed infection cases and age-specific case-fatality rates. We use data on diagnosed COVID-19 cases and death counts attributable to COVID-19 by age for China, Germany, Italy, South Korea, Spain, the United States, and New York City. We calculate the CFR for each population at the latest data point and also for Italy, Germany, Spain, and New York City over time. We use demographic decomposition to break the difference between CFRs into unique contributions arising from the age-structure of confirmed cases and the age-specific case-fatality. In late June 2020, CFRs varied from 2.2% in South Korea to 14.0% in Italy. The age-structure of detected cases often explains more than two-thirds of cross-country variation in the CFR. In Italy, the CFR increased from 4.2% to 14.0% between March 9 and June 30, 2020, and more than 90% of the change was due to increasing age-specific case-fatality rates. The importance of the age-structure of confirmed cases likely reflects several factors, including different testing regimes and differences in transmission trajectories; while increasing age-specific case-fatality rates in Italy could indicate other factors, such as the worsening health outcomes of those infected with COVID-19. Our findings lend support to recommendations for data to be disaggregated by age, and potentially other variables, to facilitate a better understanding of population-level differences in CFRs. They also show the need for well-designed seroprevalence studies to ascertain the extent to which differences in testing regimes drive differences in the age-structure of detected cases.

**Funding:** AvR received funding from the European Research Council (Grant # 716323; https://erc.europa.eu/). EA was financially supported by the Social Sciences and Humanities Research Council (https://www.sshrc-crsh.gc.ca) and the Fonds de recherche du Québec – Société et culture (http://www.scientifique-en-chef.gouv.qc.ca/le-scientifique-en-chef/les-fonds-de-recherche-du-quebec/). The funders had no role in study design, data collection and analysis, decision to publish, or preparation of the manuscript. All other authors received no specific funding for this work.

**Competing interests:** The authors have declared that no competing interests exist.

# Introduction

The novel Coronavirus disease 2019 (COVID-19), caused by severe acute respiratory syndrome coronavirus 2 (SARS-CoV-2), has been spreading rapidly across the world, and on March 11 2020 was recognized as a pandemic by the World Health Organization.

COVID-19 outbreaks went along with mostly regular patterns of logarithmic increase of the number of confirmed cases, with a few notable exceptions. The number of deaths associated with COVID-19, however, have evolved considerably less regularly, and case-fatality rates (CFRs) differ substantially between countries [1, 2].

Examples of this discrepancy are shown in Fig 1. As of June 30, 2020, Germany had a total of around 195 thousand confirmed infections and 9 thousand deaths, resulting in a CFR of around 4.6%. Italy, on the other hand, up to the same day, had 240 thousand confirmed cases of infection, around 34 thousand deaths, and a CFR of 14.0%. On April 13, Italy had roughly the same number of cases as Germany on April 28, and a CFR of 12.9%. Thus, the outbreak in Italy is going along with a much higher CFR, which has also increased over time [2, 3]. Also shown in Fig 1 are trends for Spain (until May 21) and New York City (until June 30), which fall somewhere between Germany and Italy.

Differences in the CFR could indicate that the risk of dying of COVID-19 among detected cases differs between countries or changes within a population over time. On the other hand, it could also imply compositional differences in the detected infections [1, 3]. Specifically, the risk of dying of COVID-19 is well-documented to increase with age. Thus, if the population of confirmed infected individuals is older in one country or time period than in another, the CFR will be higher, even if the age-specific risk of dying is the same.

Indeed, demographers have argued that age structure matters, and the age composition of the reported cases has been suggested as a potential explanation for differences in CFRs [1–5]. So far, however, there have been no assessments of the importance of the age structure of diagnosed cases versus the age-specific CFR.

In this paper, we analyze cross-country differences in observed CFRs and within-country time trends in CFRs. We use recent data on China, Germany, Italy, South Korea, Spain, the United States, and New York City to study cross-country differences, and we provide results

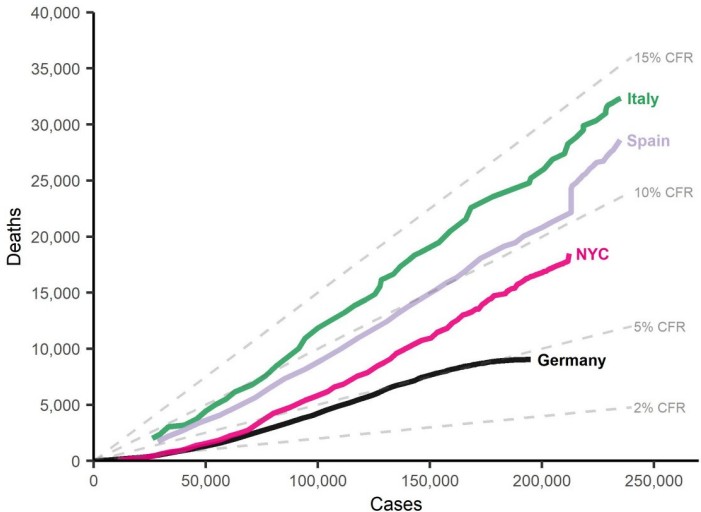

**Fig 1. COVID-19 confirmed cases and deaths, and implied case-fatality rates (CFR) in Italy (since March 9, 2020), Germany (since March 1, 2020), Spain (since March 21, 2020), and New York City (since March 22, 2020).**

on within-country time trends in Italy, Germany, Spain, and New York City. We use a standard demographic decomposition technique to disentangle two potential drivers of differences and trends: (1) the age structure of confirmed cases and (2) age-specific case-fatality rates [6].

We interpret our findings in light of the unfolding knowledge about data-driven biases affecting CFRs. Counts of confirmed cases and deaths might not be comparable across countries because of differences in case and death definitions; differences in the underestimation of cases and in their age structure as a consequence of the country-specific testing regime; reporting delays of case counts and death counts; and differences in delays between symptoms and death [1, 2]. These data-related issues might lead to over- or under-estimation of CFRs throughout the epidemic, and more reliable estimates will only be available after its conclusion. Currently, adjusting CFRs for all of these potential biases is challenging and beyond the scope of this paper. Nevertheless, the method described in this paper is also readily applicable to adjusted estimates of CFRs once they become available.

Decomposition approaches like the one used in this paper are commonly used to explain the role of age structure on changing incidence rates [7]. They have also been applied to differences in cancer fatality rates across regions with varying age structures [8]. We are not aware of any application to CFRs of infectious diseases in general and the COVID-19 pandemic in particular.

To facilitate the application of the approach described in this paper, we provide code and reproducibility materials for the open source statistical software R in a freely-accessible repository on the Open Science Framework: https://osf.io/vdgwt/. Moreover, we also provide some examples in an Excel spreadsheet in the same repository.

## Data and methods

### Data

We gathered data on the cumulative number of diagnosed infections and deaths attributable to COVID-19 for the following populations (in alphabetical order): China, Germany, Italy, South Korea, Spain, the United States, and New York City. An overview of the data is given in Table 1. For Italy, we cover the cumulative course of the epidemic over 8 periods in the analysis, starting on March 9 and ending on June 30. For all other populations, we use the data for the end of June. Results over time for Germany, Spain, and New York City are provided in the S1 File. For China, the most recent available age-specific data is from February, and for Spain it is cumulative to May 21. To provide additional context, also the cumulative number of tests

**Table 1. Populations covered in the analysis, and their cumulative detected cases, deaths, and number of performed tests.**

| Country | Date(s) | Detected cases (cumulative) | Deaths (cumulative) | Tests performed (cumulative) |
|---|---|---|---|---|
| China | February 11 2020 | 44,672 | 1,017 | - |
| Germany | June 30 2020 | 194,983 | 9,051 | 5,881,908 |
| Italy | March 9 -June 30 2020 | 240,455 | 33,736 | 5,390,110 |
| South Korea | June 30 2020 | 12,800 | 282 | 1,252,957 |
| Spain | May 21 2020 | 234,824 | 28,628 | 2,221,497 |
| United States | June 27 2020 | 2,504,175 | 119,016 | 30,446,284 |
| New York City (US) | June 30 2020 | 212,072 | 18,492 | - |

The age-specific data for China does not account for the retrospective correction of the number of deaths. The cumulative cases and deaths shown for Italy in this table are for June 30. For Germany, the number of total tests performed is from June 28, and thus from a slightly earlier date than the numbers of cases and deaths. For South Korea, the number of individuals tested is shown; i.e., the number without counting multiple tests for the same person.

for COVID-19 performed is shown in Table 1 [9], as the number of detected cases will depend on the testing regime.

All data is provided by the respective health authorities, and is collected as part of the COVerAGE database project which gathers and standardizes age-specific data on the COVID-19 pandemic [10]. The COVerAGE database is continuously updated and freely available online, but we also provide snapshots of the data used for the calculations in this paper together with the code. A complete list of sources is provided in the documentation of the database project [10].

For some of the countries (Germany, Italy, Spain, United States, and New York City) age is not available for some confirmed cases or deaths. The COVerAGE database project imputed the missing age using the observed age distribution of cases or deaths, respectively [10]. Removing these cases from the analysis altogether would have no substantive impact on the results for Germany, Italy, and New York City, as the age is only missing for few cases and deaths (less than 0.5 percent in all three cases). For Spain, however, where around 28% of cases and 44% of deaths have no recorded age, ignoring cases and deaths of unknown age would deflate age-specific case-fatality rates. For the U.S., for roughly 15% of confirmed cases the age is unknown, but age recording is relatively complete for deaths. As the age distribution of cases and deaths with unknown age might differ from those for which the age is known the imputation approach we use could potentially bias the results for Spain and the U.S. Currently, there is no indication that this is actually the case; nevertheless, the results for Spain and the U.S. need to be interpreted with more caution.

The original data is provided in different age groupings. For the decomposition, the age groups have to match. The COVerAGE database provides adjusted counts so that all countries conform with the age groups of South Korea, for which the age groups are 10-year age groups from birth to 80+. Specifically, counts were split using a recently proposed method tailored for this data situation [10, 11], based on the assumption that the age distributions of case and death rates are smooth; i.e., that there are no discontinuities or abrupt changes in rates over age. The S1 File show the original age groups of the data.

## Case-fatality rates

The COVID-19 case-fatality rate (CFR) is defined as the ratio of deaths (D) associated with COVID-19 divided by the number of detected COVID-19 cases (N): CFR = D / N. In our application, the death and case counts are cumulative counts up to a certain date.

If case counts and death counts are available by age, which is our situation, the CFR can also be written as a sum of age-specific CFRs weighted by the proportion of cases in a certain age group. We use $a$ as an index to denote different age groups. These age groups could, for instance, be 0 to 9 years, 10 to 19 years, and so on, but other groupings are also possible. We define age-specific CFRs as $C_a = \frac{D_a}{N_a}$; i.e., the number of deaths in age group $a$ divided by the number of cases in the same age group. The proportion of cases in age group $a$ is given by $P_a = \frac{N_a}{N}$. Using this notation, the CFR can be written as a weighted average of age-specific CFRs:

$$CFR = \sum P_a C_a.$$

We use the weighted expression and a mathematical decomposition approach introduced by Kitagawa to separate the difference between two CFRs into two distinct parts, one attributable to age-structure of cases and another to age-specific case-fatality [6]. The method attributes the total difference into these two components, leaving no residual. In other words, if we

use *i* and *j* to index two different populations, then the decomposition approach splits the difference between their CFRs into

$$CFR_i - CFR_j = \alpha + \delta,$$

where the $\alpha$-component captures the effect of the age structure of cases, and the $\delta$-component indicates the part of the difference attributable to age-specific case-fatality. The details of the method are described in the S1 File, which also provides a step-by-step walk-through of the decomposition and its interpretation.

## Results

### Country comparisons

Table 2 shows results for cross-country comparisons using the data from South Korea (June 30) as a reference, with countries sorted by increasing CFR. We chose South Korea as the reference because its CFR is arguably the closest match to its actual infection rate due to extensive testing relative to the number of confirmed cases and an earlier onset of the epidemic; moreover, the CFR was comparably low, and decompositions will estimate what factor leads other countries to differ from this low CFR setting, making results easy to interpret. For all other countries, we also use June 30 or the closest date available to us, as shown in Table 1. In the S1 File, we provide additional results using Germany (low CFR) and Italy (high CFR) as reference countries.

Based on the cumulative data up to June 30, South Korea had a CFR of 2.2% (first line of the table, column "CFR"). For all countries the difference to the South Korean CFR is shown in the third column of the table (South Korea minus the respective country). The fourth and fifth column of the table show the absolute contributions of the case age distribution and age-specific fatality components, respectively. A negative number for the age structure indicates an older age structure of detected cases compared to South Korea, while a negative number for the fatality component indicates higher age-specific case-fatality rates compared to South Korea. The sixth and seventh column of the table indicate the relative contributions of the components.

All countries and regions have a higher CFR than South Korea, as indicated by the negative difference shown in column four of Table 2, and some of the differences are substantial. For instance, the Italian CFR is almost seven times as high.

**Table 2. Results of the cross-country decompositions of case-fatality rates (CFRs) using South Korea as a reference case.**

| Country (1) | CFR (2) | Difference (3) | Age ($\alpha$) component (4) | Fatality ($\delta$) component (5) | Age ($\alpha$) component, relative (6) | Fatality ($\delta$) component, relative (7) |
|---|---|---|---|---|---|---|
| South Korea | 0.022 | | | (Reference) | | |
| China | 0.023 | -0.001 | -0.002 | 0.001 | 66.3% | 33.7% |
| Germany | 0.046 | -0.024 | -0.018 | -0.006 | 74.7% | 25.3% |
| USA | 0.048 | -0.025 | -0.011 | -0.014 | 43.6% | 56.4% |
| New York City | 0.087 | -0.065 | -0.015 | -0.050 | 23.4% | 76.6% |
| Spain | 0.122 | -0.100 | -0.070 | -0.030 | 70.1% | 29.9% |
| Italy | 0.140 | -0.118 | -0.077 | -0.041 | 65.3% | 34.7% |

The third column shows the difference between each country and South Korea, and is calculated as the CFR of South Korea minus the CFR of the respective country. Data for all countries is for June 30, except China (February 11), Spain (May 21), and the United States (June 27).

**Table 3. Development of the Italian case-fatality rate (CFR) over time.**

| Date (1) | CFR (2) | Difference (3) | Age ($\alpha$) component (4) | Fatality ($\delta$) component (5) | Age ($\alpha$) component, relative (6) | Fatality ($\delta$) component, relative (7) |
|---|---|---|---|---|---|---|
| 09 March 2020 | 0.043 | | | (Reference) | | |
| 23 March 2020 | 0.087 | 0.044 | -0.005 | 0.048 | 8.55% | 91.45% |
| 2 April 2020 | 0.118 | 0.075 | -0.005 | 0.081 | 6.33% | 93.67% |
| 16 April 2020 | 0.126 | 0.083 | -0.003 | 0.085 | 2.91% | 97.09% |
| 7 May 2020 | 0.131 | 0.088 | 0.000 | 0.088 | 0.02% | 99.98% |
| 26 May 2020 | 0.137 | 0.094 | -0.001 | 0.095 | 0.56% | 99.44% |
| 16 June 2020 | 0.139 | 0.097 | -0.001 | 0.098 | 0.99% | 99.01% |
| 30 June 2020 | 0.140 | 0.098 | -0.001 | 0.099 | 1.25% | 98.75% |

The third column gives the difference between the CFR of the respective date minus the CFR of March 9.

In many cases, the (relative) contributions of the $\alpha$-component (age structure) seem to be larger than the $\delta$-component (fatality), and the $\alpha$-component is always negative. This means that the age structure of cases is an important factor in explaining why most countries we study fare worse than South Korea. For instance, in the two cases with the highest CFRs—Italy and Spain—the relative contributions were similar with the $\alpha$-component explaining around two thirds of the difference (Italy: 65.3%; Spain: 70.1%), and the $\delta$-component explaining the remainder. In Germany, the case age structure also is the main driver of the difference in CFRs relative to South Korea, and explains close to 75% of the difference. The US and New York City seem to be an exception, and the high CFR compared to South Korea seems to be mostly due to higher mortality of diagnosed individuals.

## Trends over time

For Italy we have a relatively long time series of data spanning several months. Table 3 documents how the Italian CFR evolved from March 9 to June 30, with selected dates presented in between. Similar analyses for Germany, Spain, and New York City can be found in the S1 File, and we briefly comment on the results below. The CFR of March 9 is used as a reference, and the decomposition shows which factor is driving the trend in the CFR. From the beginning to the end of the period under study the CFR tripled, from 4.3% to 14.0%. This increase over time is largely driven by worsening fatality of COVID-19 –the fatality component explaining more than 90% of the change in almost all time periods—and changes in the age structure of cases only played a minor role, with detected cases moving to a more favorable (younger) age distribution and slightly counteracting the effect of worsening fatality. As a robustness check we changed the reference period from March 9 to March 21 (CFR: 8.1%). This again resulted in the fatality component explaining more than 90% of the change in CFR. The results for Spain and New York City in the S1 File show that for these populations the increases in the CFR were also mostly driven by worsening fatality, although to a lesser extent than in Italy. In contrast, in Germany the case age component almost explained 99% of the more than twofold increase in CFR between March 21 (CFR: 1.8%) and June 30 (CFR: 4.6%).

## Discussion

Case-fatality rates (CFRs) associated with COVID-19 vary strongly across countries and over time within countries. Our findings show that there is substantial variation in which factor

explains the differences in CFRs. Differences in the age distribution of detected infections in some cases explain a substantial part of the total difference in CFRs. In particular, in many cases more than 50% of the difference in CFRs between countries with a low CFR and a high CFR can be explained by the age structure of detected infections. In contrast, in Italy, we observe a substantial increase in the CFR over time, mostly attributable to increasing age-specific case-fatality.

Ultimately, the approach discussed here does not directly explain why the age structure of confirmed cases or the age-specific case-fatality rates matter more in one case and less in another, and some expertise about the contexts which are being compared is required to interpret results. We discuss some potential explanations below, including potential data-related issues and biases.

Differences in the age structure of the populations which are being compared are unlikely to be a major driver of the age component that we estimated here, as the age composition of confirmed cases does not necessarily match the age composition of the population. For instance, according to Eurostat, the proportion of the population aged 80+ in 2019 was 7% in Italy and 6.5% in Germany, while in our data the proportion of reported infections in the same age range was 25% for Italy and only 11% in Germany.

Differences in testing regimes are a plausible mechanism driving both the different age structures of detected cases, as well as different age-specific fatality rates to the extent that denominators are underestimated in distinct degrees [3, 12, 13]. Results not shown here indicate that early in the pandemic in March the difference in the CFRs of South Korea and Germany—two countries with extensive and early tracing and testing of contacts of known cases—was largely driven by differences in fatality. The low contribution of the case age distribution component to the CFR disparity between South Korea and Germany suggests that these countries might have been more successful at catching the mild and asymptomatic cases among the younger population groups. Since then, the CFR of Germany has increased and the age structure of confirmed cases has shifted to higher ages, and the age structure has become more important in explaining the gap between South Korea and Germany, making test numbers alone an unlikely explanation for the different age structure of detected cases.

Differences in the COVID-19 transmission pathways might also be a factor. Depending on contact patterns and household structure, the elderly population might be affected earlier in some countries than in others, leading to a less favorable age distribution of infections [4, 14]. This could be relevant in explaining why the age distribution plays such a large role for the two countries with by far the highest CFR, Spain and Italy, which have a relatively large proportion of individuals living with their elderly parents or grandparents, and comparatively intensive intergenerational contact [15–18].

Disparities in age-specific case-fatality rates across countries may result from differences in age-specific prevalence of comorbidities, which exacerbate the risk of death from COVID-19 considerably [1, 19] or differences in quality or saturation levels of the healthcare system, among other potential factors [20]. The trend over time in the Italian CFR is an example where changes in age-specific case-fatality rates are driving trends, instead of changes in the case age distribution. This likely reflects the worsening situation in Italy over time as its health care system got under increasing pressure [12, 21]. However, an increase in CFR could also be expected once containment measures become effective, and newly confirmed cases increase at a slower pace than deaths from cases acquired prior to containment policies.

Only once an epidemic reaches its final conclusion and all cases have either resulted in recovery or fatalities, can the importance of the age difference in cases on CFRs be assessed with an acceptable degree of accuracy [22]. In this context a distinction should be made between CFRs, which are solely based on detected cases, and infection fatality rates (IFRs),

which estimate the risk of dying once infected, including both confirmed and undiagnosed cases. Ideally policies for containing the spread of a virus would be designed on the basis of IFRs. However, particularly early on in an epidemic, the CFR is the only metric available until the extent of known data-driven biases can be assessed [1, 12, 13, 23–26].

Data quality can affect both the age composition of detected cases and age-specific case-fatality rates. For instance, counts may be affected by issues like reporting delays or censoring, or by inconsistent case definitions [1, 2, 23, 24, 27]. In many countries, only deaths occurring in hospitals are being reported in a timely manner [28], underestimating the full death count which would include deaths at home and in institutions. Deaths may be underestimated because of a lack of testing both before and after death. Countries might also differ in how they code deaths from underlying or contributory causes [28]. Excess all-cause mortality compared to a seasonal all-cause mortality baseline are suggestive that there is currently considerable underreporting of COVID-19 deaths, even if some of these deaths might be related to delayed or avoided medical treatment from other causes of death [29].

The relative importance of both the case age structure and mortality components could also be affected by comparing countries at different stages of the epidemic. This could result from cases not being detected at the beginning of the epidemic, or from differences in the lag between infection and death [12, 26, 30]. Generally, CFRs are highest at the beginning of an infectious outbreak, when the most serious cases are the most readily detected, and declines as testing capacity increases and less serious cases are identified [26]. This has notably not been the case for the COVID-19 epidemic, where the CFR has generally been increasing. Likely this reflects the success of widespread containment measures enacted in response to increasing caseloads. Newly identified cases are increasing more slowly than deaths, despite increases in testing capacity.

The application of the method we present in this paper is not limited to decomposing the current estimates of CFRs. It can also be applied to CFR estimates which have been corrected for biases, and to IFRs. It can, in principle, also be applied to excess all-cause weekly mortality counts, although this is not without challenges; we provide more discussion and some exploratory results on decomposing differences in excess mortality in the S1 File. Thus, while the data currently available as input for the decomposition approach might be of varying quality, this is not a flaw of the method itself. As data quality improves over time and adjustment methods become available our approach will continue to provide insights into differences and trends in mortality associated with COVID-19.

Finally, the choice of age groups may have affected our results. If ages were grouped too widely it might hide actual age-specific case-fatality differences. For instance, if the median age within the 10-year aggregated age groups that we used differed between populations, this would reduce the case-age structure explanation and inflate the age-specific mortality explanation. Finally, there are alternative decomposition techniques that might yield different results. However, differences are expected to be rather small; indeed, applying the method of Horiuchi and colleagues [31] to our data yields virtually the same results (results available upon request).

The results of this study add weight to recommendations for data to be disaggregated by age and potentially other variables to facilitate a better understanding of population-level differences in CFRs. Equally important will be well designed seroprevalence studies to ascertain the extent to which our findings are driven by differences in testing regimes, particularly in the diagnosis of mild and asymptomatic cases. To this extent we are encouraged by the recent start of such a study in Germany in line with official WHO recommendations [32, 33] and by first results from a large, population-representative studies from Italy and Spain [34, 35].

Overall, our results show that differences between countries with low and high CFRs can be driven to a significant extent by the age structure of confirmed cases. Decomposing differences in case-fatality rates over time or between countries reveals important insights for monitoring the spread of COVID-19. An accurate assessment of these differences in CFR across countries and over time are crucial to inform and determine appropriate containment and mitigation interventions, such as social confinement and mobility restrictions.

## Supporting information

**S1 File. Details on methods and additional results.**
(DOCX)

## Acknowledgments

We thank Catalina Torres for collecting parts of the data and for helpful comments and suggestions.

## Author Contributions

**Conceptualization:** Christian Dudel, Tim Riffe, Enrique Acosta, Alyson van Raalte, Mikko Myrskylä.

**Data curation:** Christian Dudel, Tim Riffe, Enrique Acosta, Cosmo Strozza.

**Formal analysis:** Christian Dudel, Tim Riffe, Enrique Acosta.

**Investigation:** Christian Dudel, Tim Riffe, Enrique Acosta.

**Methodology:** Christian Dudel.

**Project administration:** Christian Dudel.

**Software:** Christian Dudel, Tim Riffe, Enrique Acosta.

**Visualization:** Enrique Acosta.

**Writing – original draft:** Christian Dudel.

**Writing – review & editing:** Christian Dudel, Tim Riffe, Enrique Acosta, Alyson van Raalte, Cosmo Strozza, Mikko Myrskylä.

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
