## [Decision Letter · Decision Letter 0]

8 Jul 2020

PONE-D-20-12360

Monitoring trends and differences in COVID-19 case-fatality rates using decomposition methods: Contributions of age structure and age-specific fatality

PLOS ONE

Dear Dr. Dudel,

Thank you for submitting your manuscript to PLOS ONE. After careful consideration, we feel that it has merit but does not fully meet PLOS ONE’s publication criteria as it currently stands. Therefore, we invite you to submit a revised version of the manuscript that addresses the points raised during the review process.

As I mentioned by e-mail, it was quite difficult to secure reviewers for this article during this very busy period. Many candidate reviewers declined my invitation due to lack of time or conflicts of interest. Two reviewers have commented on your work, and I am very grateful for their dedication to read and carefully review your mauscript. Based on these reviews and my reading of the paper, I would like to open up the possibility of revising the article to strengthen it, taking care to answer the questions of the second reviewer, in particular her concern about the use of COVID notified death data, which represents only a portion of COVID-related deaths. I think that this point can be addressed through expanding the discussion or ideally through applying this method on weekly mortality death counts available in the HMD. I also agree with reviewer 2 about her suggestion to supplement the trends over time with other countries (the US is a candidate to consider). Could you also consider an update of the results to capture the decline in some countries and consequently a re-reading of some elements of the conclusion? The paragraph on seroprevalence surveys should also be updated in the light of the new studies available.

We look forward to receiving your revised manuscript.

Kind regards,

Bruno Masquelier, PhD

Academic Editor

PLOS ONE

Journal Requirements:

Reviewers' comments:

Reviewer's Responses to Questions

**Comments to the Author**

1. Is the manuscript technically sound, and do the data support the conclusions?

Reviewer #1: Yes

Reviewer #2: Partly

2. Has the statistical analysis been performed appropriately and rigorously? 

Reviewer #1: Yes

Reviewer #2: Yes

3. Have the authors made all data underlying the findings in their manuscript fully available?

Reviewer #1: Yes

Reviewer #2: Yes

4. Is the manuscript presented in an intelligible fashion and written in standard English?

Reviewer #1: Yes

Reviewer #2: Yes

5. Review Comments to the Author

Reviewer #1: This paper makes an important contribution by highlighting the association between population-level case fatality rates (CFRs) for COVID-19 and the age-structure of underlying infections as well as age-specific case fatality rates. The authors show that the age structure of confirmed cases can explain up to 2/3 of the differences in CFRs across countries.

The authors are to be commended for their demonstrated commitment to open science in making the data and code freely available, especially during the quickly changing landscape of research during the COVID-19 pandemic. The tool that is provided will be extremely valuable to scholars working to compare CFRs and the severity of COVID-19 across countries and time. Given the timely nature of the work and the clearly written contribution, I recommend publication as is.

One point of clarification: Could the authors clarify in the text if you also use data over time for Germany as in Figure 1.

Reviewer #2: This paper proposes an interesting method to compare and understand the impact of COVID19 on different countries. While this paper addresses an important and current topic with an intuitive and sound method, I found the analysis is not build upon robust data.

The authors use a mathematical decomposition method to separate the difference between two crude fatality rates (CFR) into an age-structure part and an age-specific case-fatality rate part, leaving no residual. They provide cross-country comparisons but also within-country comparisons over time. Countries included are China, Germany, Italy, South Korea, Spain, United States in addition to the city of New York. The method requires selecting a reference country (South Korea) or a reference date (9 March for Italy) to which they compare other CFRs. All included countries had a higher CFR than South Korea. According to the authors, age structure has the biggest relative importance in explaining the difference in CFRs with South Korea. The CFR of Italy has been decomposed over time and the authors highlight that the fatality component is the one driving the evolution of the CFR.

Strengths:

- The decomposition method is intuitive and well explained. In addition, it is proposed as a pertinent method to decompose the difference between other metrics (IFR, excess all-cause).

- Authors provide explanations for differences in CFR according to two important demographical factors.

- Data is subject to important limitations but these limitations are clearly highlighted in Introduction and are well discussed in Discussion.

- They reference the available literature which shed more lights on the topics and enrich the discussion.

Weaknesses:

- The variable used in their analysis is the ratio between reported COVID19 deaths and reported cases. Reported cases is highly determined by testing policies of a given country. Concerning reported COVID19 deaths, there is now a consensus in the demographic research community that the impact of COVID19 should be studied with all-cause death data. Reported COVID19 deaths are subject to multiple biases (national definition of COVID19 deaths, nurse houses not always included, reporting delays, ..). Having this in mind, it is hard to draw any robust conclusions from the results.

- Each country’s reported deaths might be subject to different biases. The fact that the method relies on the comparison to a reference country makes it even harder to understand in which direction biases go.

- For a majority of the analyzed country, the end date of the analysis does not consist of the end of the first wave. Thus, even when focusing only on the first wave, only part of the picture is studied.

Areas for improvement/Questions to the authors:

Major:

1) p.15, first paragraph: The authors mention that the method could be used on excess all-cause weekly mortality counts. This data is now available for some countries (see the Human Mortality Database for example). I would suggest that the authors use excess deaths that are less subject to bias and compare their results.

2) p.10: Trends over time were only studied for Italy. I would suggest the authors to analyze how trends differ between countries (i.e with Spain, the US) or at least address why they don’t.

3) The authors should present data on the amount of testing performed by each country, if possible, over the available age groups (and over time for Italy). It would be more transparent to read results having this information on the side.

4) p.4: “Counts of confirmed cases and deaths might not be comparable across countries because of differences in case and death definitions”. Despite rightly highlighting this fact, the purpose of the selected method is cross-country comparisons. Having this in mind, can the results support any conclusion?

5) Amount of missing data is only presented for Spain. What is the amount of missing data in the other countries? Could you add this information for all the countries?

6) p.6: “The database project imputed the missing age using the observed age distribution of cases or deaths, respectively”. Doesn’t this method exacerbate the bias if some ages are rarely reported (i.e deaths in nurse houses)?

Minor:

7) p.7 “Counts were split using a recently proposed method tailored for this data situation [9,10]”. For transparency, I would recommend to highlight assumptions required by this method (such as the underlying smoothness assumption of the counts).

8) In order to provide more descriptive statistics, could figure 1 be done for all the countries included in the analysis?

6. PLOS authors have the option to publish the peer review history of their article (what does this mean?). If published, this will include your full peer review and any attached files.

Reviewer #1: No

Reviewer #2: No

---

## [Author Response · Author response to Decision Letter 0]

10 Aug 2020

For our replies to the reviewer and editor comments please see the attached documents.

---

## [Decision Letter · Decision Letter 1]

27 Aug 2020

Monitoring trends and differences in COVID-19 case-fatality rates using decomposition methods: Contributions of age structure and age-specific fatality

PONE-D-20-12360R1

Dear Dr. Dudel,

We’re pleased to inform you that your manuscript has been judged scientifically suitable for publication and will be formally accepted for publication once it meets all outstanding technical requirements.

Kind regards,

Bruno Masquelier, PhD

Academic Editor

PLOS ONE

Additional Editor Comments (optional):

Reviewers' comments:

Reviewer's Responses to Questions

**Comments to the Author**

1. If the authors have adequately addressed your comments raised in a previous round of review and you feel that this manuscript is now acceptable for publication, you may indicate that here to bypass the “Comments to the Author” section, enter your conflict of interest statement in the “Confidential to Editor” section, and submit your "Accept" recommendation.

Reviewer #2: All comments have been addressed

2. Is the manuscript technically sound, and do the data support the conclusions?

Reviewer #2: Yes

3. Has the statistical analysis been performed appropriately and rigorously? 

Reviewer #2: Yes

4. Have the authors made all data underlying the findings in their manuscript fully available?

Reviewer #2: Yes

5. Is the manuscript presented in an intelligible fashion and written in standard English?

Reviewer #2: Yes

6. Review Comments to the Author

Reviewer #2: This paper proposes an interesting method to compare and understand the impact of COVID19 on different countries. The paper addresses an important and current topic with an intuitive and sound method. The revision allows to contextualise better the different important results and improves transparency.

Thank you for addressing rigorously my concerns.

7. PLOS authors have the option to publish the peer review history of their article (what does this mean?). If published, this will include your full peer review and any attached files.

Reviewer #2: No

---

## [Editor Report · Acceptance letter]

1 Sep 2020

PONE-D-20-12360R1 

Monitoring trends and differences in COVID-19 case-fatality rates using decomposition methods: Contributions of age structure and age-specific fatality 

Dear Dr. Dudel:

I'm pleased to inform you that your manuscript has been deemed suitable for publication in PLOS ONE. Congratulations! Your manuscript is now with our production department. 

Kind regards, 

on behalf of

Dr. Bruno Masquelier 

%CORR_ED_EDITOR_ROLE%

PLOS ONE